# Dose Reduction to the Swallowing Apparatus and the Salivary Glands by De-Intensification of Postoperative Radiotherapy in Patients with Head and Neck Cancer: First (Treatment Planning) Results of the Prospective Multicenter DIREKHT Trial

**DOI:** 10.3390/cancers12030538

**Published:** 2020-02-26

**Authors:** Marlen Haderlein, Stefan Speer, Oliver Ott, Sebastian Lettmaier, Markus Hecht, Sabine Semrau, Benjamin Frey, Claudia Scherl, Heinrich Iro, Marco Kesting, Rainer Fietkau

**Affiliations:** 1Department of Radiation Oncology, Universitätsklinikum Erlangen, Friedrich-Alexander-Universität Erlangen-Nürnberg (FAU), 91054 Erlangen, Germany; stefan.speer@uk-erlangen.de (S.S.); oliver.ott@uk-erlangen.de (O.O.); sebastian.lettmaier@uk-erlangen.de (S.L.); markus.Hecht@uk-erlangen.de (M.H.); sabine.semrau@uk-erlangen.de (S.S.); benjamin.frey@uk-erlangen.de (B.F.); rainer.fietkau@uk-erlangen.de (R.F.); 2Department of Otorhinolaryngology, Universitätsklinikum Mannheim, Medical Faculty Mannheim, University of Heidelberg, 68167 Mannheim, Germany; claudia.scherl@umm.de; 3Department of Otorhinolaryngology, Universitätsklinikum, Friedrich-Alexander-Universität Erlangen-Nürnberg (FAU), 91054 Erlangen, Germany; heinrich.iro@uk-erlangen.de; 4Department of Oral and Maxillofacial Surgery, Universitätsklinikum, Friedrich-Alexander-Universität Erlangen-Nürnberg (FAU), 91054 Erlangen, Germany; marco.kesting@uk-erlangen.de

**Keywords:** de-intensification, radiotherapy, head neck cancer, postoperative, ipsilateral elective neck irradiation only, salivary glands, swallowing apparatus

## Abstract

Aim: Evaluating radiotherapy treatment plans of the prospective DIREKHT trial (ClinicalTrials.gov, NCT02528955) investigating de-intensification of radiotherapy in patients with head and neck cancer. Patients and Methods: The first 30 patients from the DIREKHT trial of the leading study centre were included in this analysis. Standard treatment plans and study treatment plans derived from the protocol were calculated for each patient. Sizes of planning target volumes (PTVs) and mean doses to organs at risk were compared using the Student’s t-test with paired samples. Results: Mean PTV3 including primary tumor region and ipsilateral elective neck up to a dose of 50 Gy in the study treatment plans was 662 mL (+/− 165 mL standard deviation (SD)) and therefore significantly smaller than those of the standard treatment plans (1166 mL (+/− 266 mL SD). In the medial and inferior constrictor muscles, cricopharyngeal muscle, glottic and supraglottic laryngeal areas, arytenoid cartilages, contralateral major salivary glands highly significant dose reductions (*p* < 0.0001) of more than 10 Gy were achieved in study treatment plan compared to standard treatment plan. Conclusion: De-intensification of radiotherapy led to smaller planning target volumes and clinical relevant dose reductions in the swallowing apparatus and in the contralateral salivary glands.

## 1. Background

To date, there is only one prospective study [1] with a small sample size and some retrospective studies [2,3] investigating the possibility of treating ipsilateral elective neck nodes only in the postoperative situation of head and neck cancer. Moreover, the primary tumor region usually is treated with a dose up to 64–66 Gy [4,5,6]. The sole study investigating dose reduction in patients with head and neck cancer undergoing postsurgical radiotherapy (RT) is Peters et al. [7] which showed that a minimal dose of 57.6 Gy should be applied even in low-risk patients. Risk classification was done by a point score leading to a possible inclusion of even patients with T4 tumors, major nerve infiltration or resection margin <5 mm in the low-risk group. 

In low-risk patient populations with head and neck cancer 5 year- locoregional control rates of over 90% have been achieved, but about 30% of the patients suffer from grade III therapy-related side-effects, like xerostomia, dysphagia and trismus leading to a reduced quality of life [8,9,10,11,12,13]. Therefore, a need for a prospective trial investigating de-intensification of radiotherapy is recognized in a clearly defined patient population with head and neck cancer in the postoperative situation.

The prospective, non-randomised, and multicenter DIREKHT-trial investigates the possibility of de-intensification of postoperative radiotherapy in a predefined low-risk patient population with head and neck cancer (ClinicalTrials.gov, NCT02528955).

Low-risk was specified as either low-risk for local recurrence in the primary tumor region or for occurrence of any contralateral neck lymph nodes. In case of low-risk for local recurrence of the primary tumor defined as pT ≤ 2, no peritumoral lymphatic vessel invasion (L0), no peritumoral perineural spread (Pn0) and a resection safety margin of ≥5 mm, irradiation dose to the primary tumor region was reduced to a total dose of 56 Gy. In patients with a genuine low risk for contralateral neck recurrence (≤p T3, ≤3 ipsilateral neck node metastases, contralateral no lymph node metastases defined as either pN0 or contralateral cN0 in case of well lateralized oral cavity or oropharyngeal carcinoma) prescribed target volume was reduced by electively irradiating only the ipsilateral neck nodes. This resulted in three protocol treatment arms (see Figure 1).

Enrollment of patients in the leading study centre started in September 2014 and is ongoing. To date (30 November 2019) eight study centres are recruiting patients and 127 patients have been included. Inclusion of 200 patients in total is planned and because of the known different radiation response there is a stratification on human papilloma virus (HPV) status and also on tumor localisation.

The aim of the present analysis is to quantify size of irradiated target volumes and to quantify dose reduction in organs at risk, especially in the swallowing apparatus and salivary glands in study treatment plans compared to non-restricted standard treatment plans. Radiation doses of 56 Gy and more are known to cause fibrosis and for quality assurance it should also be assessed if a clinical relevant dose reduction of <56 Gy in the swallowing apparatus might be achieved or if the dose constraints of the study protocol might be improved.

## 2. Material and Methods

At the timepoint of the present analysis 30 patients of the leading study center of the DIREKHT trial (ClinicalTrials.gov, NCT02528955), the Department of Radiation Oncology at Universitätsklinikum Erlangen, have been included in the trial and were available for this treatment planning study. Additionally to the protocol-based irradiation plan, for each patient a standard treatment plan was calculated for dose comparisons. 

The protocol was approved by the Human Research Protocol Office, and all patients gave signed informed consent to participate in the trial. All procedures were performed in accordance with the Helsinki Declaration.

### 2.1. Target Volumes and Dose Prescriptions

All target volumes were contoured on an intravenous contrast-enhanced planning computertomography (CT) on axial 3 mm slices with patients in supine position fixed with a thermoplastic mask system. Fusion of preoperative diagnostic imaging (usually CT, but also MRI or FDG-PET-CT) to the planning CT was done to define preoperative gross tumor volume (GTV). Additionally, surgery and pathology reports were taken into account.

According to study protocol target volumes and dose prescriptions were as follows:

Clinical target volume (CTV) 1 included the former primary tumor region or space after resection (if the criteria for dose reduction referred to in the introduction are not met) and lymph node levels with resected lymph node metastases with extracapsular spread (ECS) (in case of ECS or soft tissue deposits ≥ 3 cm the lymph node levels should have been surrounded by a 5 mm safety margin).

CTV 2 always included the former primary tumor region or space after resection, lymph node levels with resected lymph node metastases (in case of ECS or soft tissue deposits ≥3 cm the lymph node levels should be surrounded by 5 mm safety margin) and tracheostomy (either tracheostoma or former region of the tracheotomy performed during surgery).

CTV 3 additionally to CTV2 includes elective neck nodes (criteria for target volume reduction by only treating ipsilateral elective neck nodes are mentioned in the introduction section). For detailed information on contouring elective neck nodes see Appendix A.

PTV 1, 2 and 3 resulted by giving a safety margin of 3–5 mm around each CTV1, 2 and 3. Size of safety margin (3–5 mm) is defined by each participating centre according to individual setup errors.

Prescribed dose in PTV 1 was 64 Gy, in PTV 2 56 Gy and in PTV 3 50 Gy. Single fraction dose was 2 Gy. One fraction per day and 5 fractions per week were delivered.

Target volume definition and dose prescriptions of the standard treatment plan were as follows:

CTV 1: elective neck nodes on both sides of the neck, administered dose using percutaneous radiotherapy (RT): 50 Gy; CTV 2: lymph node metastases without ECS, administered cumulative dose using percutaneous RT: 56 Gy; CTV 3: primary tumor bed and lymph node metastases with ECS, administered cumulative using percutaneous RT dose: 64 Gy.

For patients being treated in Arm 1 a PTV 1 including the primary tumor region was defined. For patients being treated in Arm 2 the PTV 3 was enlarged by including the contralateral elective neck nodes and for patients being treated inArm 3 a PTV 1 was defined and PTV 3 was enlarged. PTV 2 is the same in standard and study treatment plan.

### 2.2. Organs at Risk

The following swallowing structures were delineated according to Christianen et al. [14]: superior, medial and inferior constrictor muscles (PCMs), cricopharyngeal muscle (CPM), esophagus inlet muscle (EIM), glottic/supraglottic larynx and base of the tongue (BOT). Additional contouring of the cervical esophagus (up to the superior border of the sternum), the soft palate, the oral cavity, arytenoid cartilages, the parotid glands and the submandibular glands was performed.

### 2.3. Dose Limitations and Treatment Planning

Pinnacle² version9 treatment planning system (Philips Radiation Oncology Systems, Fitchburg, WI, USA) was used for treatment planning. PTV 3 of the study and standard treatment plan was generated by using script-based planning. Detailed information on script-based planning have been published earlier [15]. All patients were treated with VMAT (volumetric modulated arc therapy) and also the standard plans were calculated for VMAT treatment.

According to study protocol the following dose limitations should have been respected:

D_max_ (maximum dose) of spine: 45 Gy; Dmax of brainstem: 50 Gy, Dmean (mean dose) of the contralateral parotid gland <26 Gy, 

D_mean_ of each swallowing structure should have been <56 Gy if coverage of target volume was not compromised.

In the standard treatment plans the dose constraints mentioned above were also routinely respected, if possible. Treatment planning was in accordance with the guidelines of the International Commission on Radiation Units and Measurements (ICRU) report 50/83 (http://jicru.oxfordjournals.org/content/10/1.toc) to deliver the prescribed doses to the PTVs while keeping doses to the organs at risk within the constraints or as low as reasonably achievable.

Target volume of all irradiated PTVs and Dmean of ipsilateral and contralateral parotid and submandibular glands as well as swallowing structures were evaluated. Furthermore we assessed the Dmean of all swallowing structures and parotid and submandibular glands with a 3 mm safety margin in all directions in order to consider daily reposition error and organ movement.

### 2.4. Statistical Analysis

IBM SPSS Version 20 (IBM, Armonk, NY, USA) was used for calculations and descriptive statistics. Mean, maximum and minimum values of each variable and related standard deviation was calculated. Data were compared using the Student’s t-test with paired samples (Study treatment plan vs. standard treatment plan). Statistical significance was assumed for *p* ≤ 0.05.

## 3. Results

For detailed patient information see Table 1.

No patient was treated in Arm 1, 23/30 (76.7%) patients in Arm 2 and 7/30 (23.3%) patients in Arm 3. Mean PTV3 (see Figure 2) in the study treatment plans was 662 mL (standard deviation: +/− 165 mL; range: 345; 926 mL) compared to 1166 mL (standard deviation: +/− 266 mL; range: 636; 1586 mL) in the standard treatment plans (*p* < 0.000).

Mean PTV 2 was 391 mL (standard deviation: +/− 120 mL; range 170; 598 mL) in the study and standard treatment plans.

Mean PTV 1 in the study treatment plans was 170 mL (standard deviation: +/− 142 mL; range: 0; 486 mL compared to 208 mL (standard deviation: +/− 112 mL; range: 69; 486 mL) (*p* = 0.036). In seven patients an additional PTV 1 was defined for the standard treatment plan with a mean size of 164 mL (range 103; 273 mL). In the other 23 patients PTV 1 of the study plan was equal to PTV 1 of the standard treatment plan. 

A significant dose reduction in organs at risk was seen in the complete swallowing apparatus and in all salivary glands, but not in the ipsilateral submandibular gland. In study treatment plans a clinical relevant reduction of mean applied dose of <56 Gy was reached in all organs at risk except in soft palate and ipsilateral submandibular gland. Mean differences in organs at risk of more than 10 Gy difference in study treatment plans compared to standard treatment plans was seen in MCM, ICM, M. cricopharyngeus, glottic and supraglottic larynx, arytenoid cartilages, contralateral submandibular and parotid gland. For detailed information see Table 2. In Table 2 it is also shown that in superior and middle constrictor muscle, in oral cavity, in base of tongue and supraglottic larynx a dose of <56 Gy is achieved in study but not in standard treatment plan.

Dose differences in organs at risk surrounded by a 3 mm safety margin in all directions provided equal results, see Appendix A. Comparing dosimetric values of patients treated in Arm 2 and patients treated in Arm 3 it is obvious that in organs at risk near or in the primary tumor region (e.g., soft palate, scm, base of tongue, oral cavity) a further dose reduction is reached by limitating applied dose in the primary tumor region to 56 Gy in Arm 3. Therefore mean dose in all organs at risk is under 56 Gy in patients treated in Arm3 (see Appendix A). Figure 3 demonstrates a dose reduction in the primary tumor region with ipsilateral elective neck irradiation.

## 4. Discussion

This plan comparison study shows a significant dose reduction for organs at risk, especially regarding the swallowing apparatus and contralateral submandibular and parotid glands by de-intensification of radiotherapy in a low-risk patient population with head and neck cancer in the postsurgical situation. Moreover this is not only a fictive planning study but a real world setting. Patients have been irradiated with innovative optimized study treatment plans. In a previous study this low-risk patient population was evaluated retrospectively [8]. Even though patients being treated with standard-of-care radiotherapy showed excellent locoregional control rates of more than 95% after 5 years and a cumulative incidence of distant metastases of less than 6%, about 13% of the patients suffered from a second malignancy 5 years after diagnosis of head and neck cancer and more than 20% of the patients suffer from a high rate of therapy induced late side effects.

Twenty-one percent of the patients showed a grade III dysphagia and about 20% a grade II xerostomia. [8] Considering the high incidence of second cancer and the severe rate of therapy induced late side effects, the current prospective trial was started for this low-risk patient population. In case of xerostomia, there is evidence for dose–volume relationships linking the dose to the major salivary glands to a dry mouth and clear dose limits exist for the parotid gland [16]. There are also several studies [17,18,19,20,21,22,23,24,25,26,27] that demonstrate a dose-volume-relationship between late dysphagia and the radiation dose delivered to specific parts of the swallowing apparatus. A correlation between post-treatment swallowing function and the radiation dose to various structures of the swallowing apparatus, such as the constrictor muscles in general [17,28], superior PCM [18,19,20,21,22,27], middle PCM [18,19,20,22,23], inferior PCM [18,24,25], CPM [25], supraglottic larynx [17,23,26,28], the larynx in general [18,24,28], oral cavity [21] and the EIM [26], esophagus [17], and soft palate [24,27] had been reported. However different swallowing structures were contoured in these studies and the definition of the corresponding anatomically morphological CT boundaries have varied across different studies. The results of these studies are discrepant regarding the relevant swallowing structures to be spared and the optimal dose needed to be delivered. In view of these discrepant results it appears likely that optimization of posttreatment swallowing function can be achieved realistically by protecting either a complete part or coherent parts of the swallowing apparatus and not a single special structure. Consequently, one could spare either cranial or caudal portions of the swallowing structures, depending on tumor location and lymph node involvement. Dose-volume-relationships vary, but it has to be assumed that mean doses of less than 60 Gy or better less than 56 Gy should be applied in the swallowing apparatus to reach better functional outcome for the patients [29]. Indeed, the dosimetric gain that was reached in the study treatment plans was surprisingly high. Large parts of the swallowing apparatus were irradiated with a mean dose of less than 56 Gy and mean dose in contralateral submandibular gland was less than 20 Gy in most patients. According to reported dose-volume-relationships [16,29] these dose reductions in study treatment plans should lead to an an improved outcome for the patients regarding late side-effects like xerostomia or dysphagia. 

Prospective randomized trials have already shown the benefit of parotid sparing IMRT [30,31] Feng et al. [32,33] showed in a non-randomized trial that dose reduction in the swallowing apparatus led to reduced late dysphagia in patients with oropharyngeal cancer after definitive radiotherapy. But results of prospective randomized trials [34] investigating swallowing sparing IMRT are still pending.

Dose reduction in study treatment plans compared to standard treatment plans was especially seen in contralateral submandibular and parotid gland, medial/inferior constrictor muscle, musculus cricopharyngeus and larynx. In case of all primary tumors being localized either in the oral cavity or oropharynx the absolute dose reduction in the oral cavity, base of tongue and soft palate is much lower.

Due to the inclusion of tracheostoma or tracheostomy in PTV 2 and PTV 3 the cervical esophagus may not be spared to the maximal extent. Therefore especially in patients with tracheostoma or tracheostomy during surgery there is no difference between study and standard treatment plan resulting in a relatively low mean difference of delivered dose comparing standard and study treatment plan. The contralateral submandibular gland was the OAR which could be spared most in experimental treatment plans. That is due to the fact that no patient was treated in Arm 1 and therefore all of the 30 patients received ipsilateral elective node irradiation only. There is evidence of a relationship between dose delivered in the contralateral submandibular gland and percutaneous endoscopic gastrostomy tube dependence after radiotherapy [35]. But there is also a percentage of patients who underwent submandibulectomy on both sides during primary tumor surgery and neck dissection. The influence of submandibulectomy and dose-volume relationships on swallowing outcome after radiotherapy will be considered in the final analysis of the trial. 

It has to be mentioned that dosimetric results might not represent those of the planned entire cohort of the ongoing trial as none of the first 30 patients included in this treatment study were treated in Arm1. Currently there are hardly any patients meeting inclusion citeria for Arm1 and that less than 10% of the included patients in the DIREKHT trial were treated in Arm 1. Only the first 30 patients of the leading study centre were included in this treatment planning study, because creating a standard treatment plan and comparing it to study treatment plan was not part of the study protocol and therefore this information was not available from the other participating centres. But in our opinion this treatment plan comparison was important for quality assurance and to find out if if the dose constraints and target volumes of the study protocol might be improved. And for answering this question we think 30 patients were adequate.

Advances in head and neck surgery have led from radical neck dissection to selective neck dissection in selected patients with squamous cell carcinoma of the oral cavity, pharynx and larynx [36]. In radiation oncology modern techniques such as IMRT or VMAT combined with image-guidance allow a higher precision in dose application and therefore individualized treatment approaches should be investigated [37,38,39,40]. The aim of the DIREKHT trial is to reduce late toxicity after postoperative radiotherapy without increasing locoregional failure. All patients included in this study receive swallowing endoscopy before radiotherapy and 6 and 12 months after the end of radiotherapy. Moreover patients answer quality of life questionnaires at fixed dates. An interim analysis was planned after the enrolment of the first 100 patients and is presently ongoing. But at the moment results of locoregional control rate and late side effects, especially xerostomia and dysphagia are still pending and it is currently unknown if smaller PTVs and lower doses in the swallowing apparatus and contralateral submandibular and parotid gland lead to less long-term side-effects and better quality of life. It is important to bear in mind that in this analysis only a small number of patients has been included and this might influence statistical observation. Moreover there are more factors influencing long-term dysphagia in patients with head and neck cancer, e.g., smoking [24,25], primary tumor site [18,21,23], T-stage [18,23,41], alcohol consumption [27] and pretreatment dysphagia [18,23,41]. These factors and even more patient-, tumor and treatment-related parameters will be considered in the final analysis of the study.

## 5. Conclusions

De-intensification of radiotherapy in a pre-defined low-risk patient population with head and neck cancer leads to significant smaller planning target volumes and significant lower doses in parts of the swallowing apparatus and the contralateral salivary glands.

## Figures and Tables

**Figure 1 cancers-12-00538-f001:**
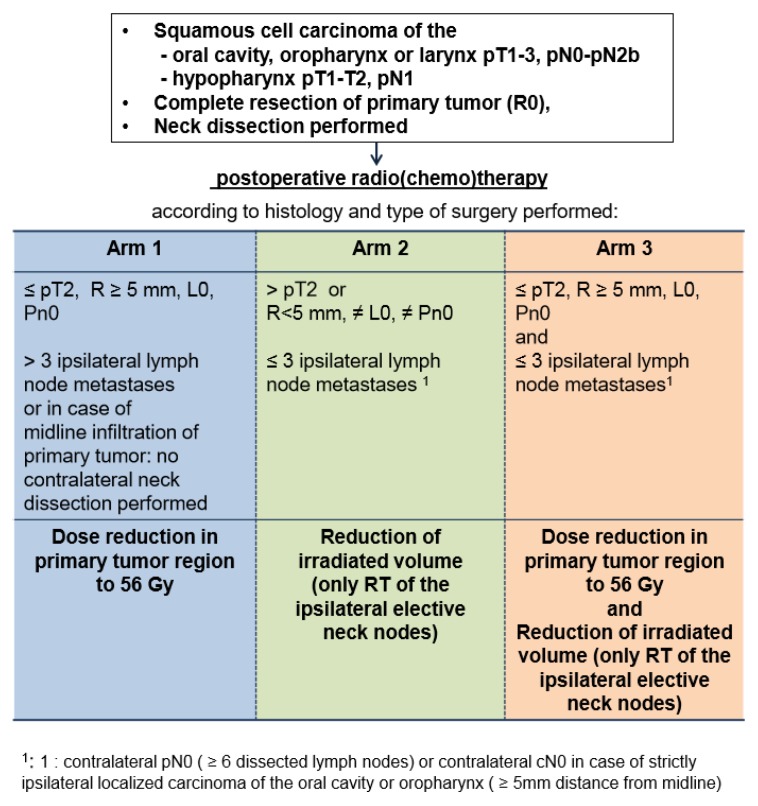
Flow Chart of the DIREKHT trial.

**Figure 2 cancers-12-00538-f002:**
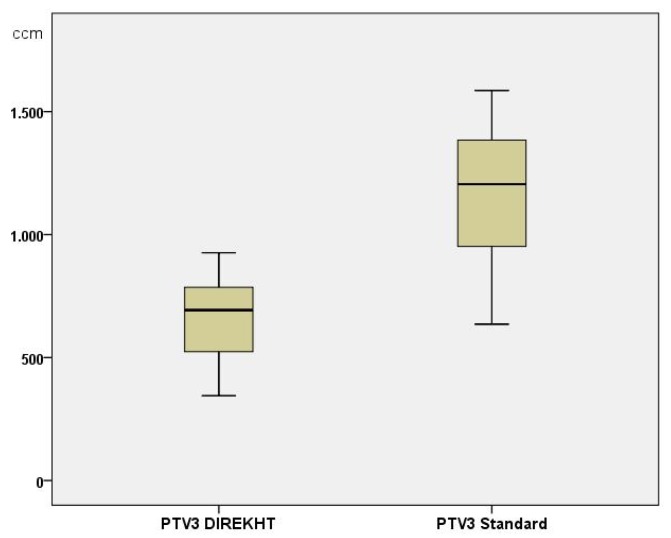
Planning target volume 3 (PTV3) of study treatment plan compared to standard treatment plan.

**Figure 3 cancers-12-00538-f003:**
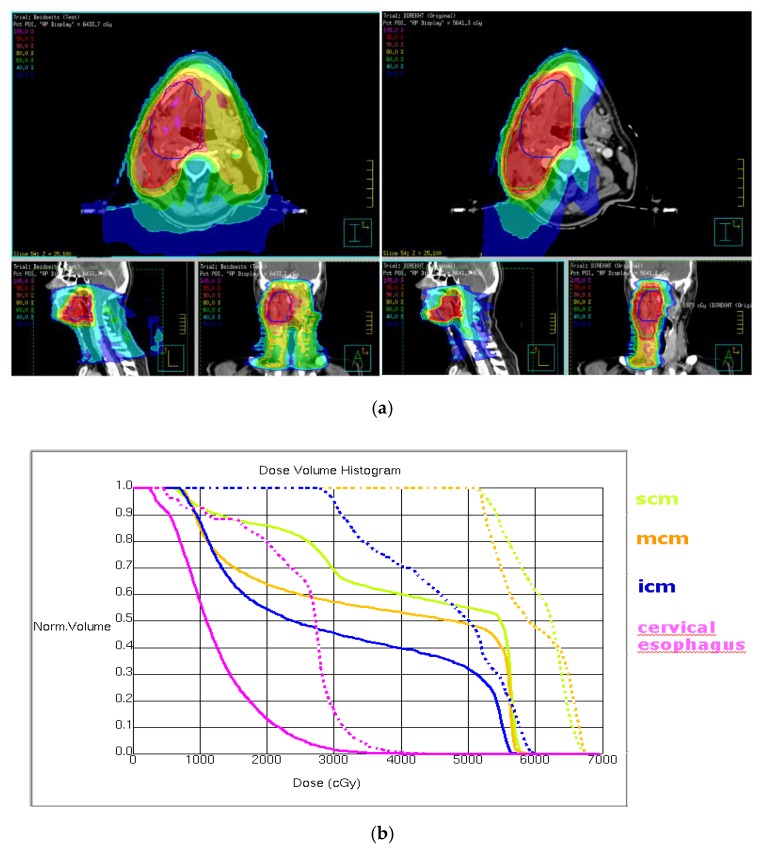
Male patient with a HPV-associated tonsillar carcinoma on the right side; TNM according to 7th edition of TNM classification: pT1pN2b (3/29 without ECS) L0 V0 Pn0 R0 cM0. According to study protocol this patient was treated with a reduced radiotherapy dose of 56 Gy in the primary tumor region only with ipsilateral elective neck irradiation. In the standard treatment plan, the patient would have received a total dose of 64 Gy in the primary tumor region and bilateral elective neck irradiation. (**a**) Dose distribution of the standard and study treatment plan (blue contour within the high-dose-volume is the boost which is additionally applied in the standard but not in the study treatment plan) with the study treatment plan showing especially significant reduced dose in contralateral neck and pharyngeal structures. In the study treatment plan most parts of the swallowing structures receive less than 30% of the prescribed dose while in the standard treatment plan large parts of the swallowing apparatus receive 40 to 60% of prescribed dose. (**b**) Dose-volume histogram of superior (light green), medial (orange), inferior (blue) constrictor muscle and cervical esophagus (pink) shows significantly smaller doses in constrictor muscles and cervical esophagus in the study (solid line) compared to the standard (dotted line) treatment plan.

**Table 1 cancers-12-00538-t001:** Patient characteristics.

Characteristics		No. of Patients	%
Sex			
	Male	19	63.3
	Female	11	36.7
Age at diagnosis, y			
	Median	59	
	Range	40–80	
Primary tumor site			
	Oral cavity	9	30
	Oropharynx	21	70
	Base of the tongue	4	
	Tonsil	16	
	Other	1	
pT classification			
	T1	10	33.3
	T2	16	53.3
	T3	4	13.3
pN classification			
	N0	1	3.3
	N1	9	30
	N2a	10	33.3
	N2b	10	33.3
Perinodal spread			
	Yes	6	20
	No	24	80
Lymphangiosis			
	L0	25	83.3
	L1	5	16.7
Hemangiosis			
	V0	28	93.3
	V1	2	6.7
Perineural spread			
	Pn0	24	80
	Pn1	6	20
Grading			
	G1	0	0
	G2	9	30
	G3	21	70
Tracheostomy			
	yes	5	16.7
	yes, temporary during surgery	15	50
	no	10	33.3
HPV			
	Positive	9	30
	Negative	16	53.3
	Not defined	5	16.7
Neck dissection			
	Only ipsilateral	13	43.3
	Bilateral	17	56.7
Total removed lymph nodes, n			
	Median	32	
	Range	12–90	
No. of affected lymph nodes, n			
	Median	1	
	Range	0–3	
Simultaneous Chemotherapy, n			
	Yes	15	50
	No	15	50

**Table 2 cancers-12-00538-t002:** Mean doses applied in swallowing apparatus and major salivary glands in study treatment plan compared to standard treatment plan (Organs at risk in which a mean dose of <56 Gy is applied in study but not in standard treatment plans are underlined).

Organs at Risk	DIREKHT Treatment Plan Mean Dose Applied (Gy) ± Standard Deviation	Standard Treatment Plan Mean Dose Applied (Gy) ± Standard Deviation	*p* Value	Median Difference Between Study-and Standard Treatment Plans (Gy) (Minimum; Maximum)
Superior constrictor muscle (SCM)	53.8 ± 6.4	61.3 ± 2.0	0.000	6.9 (−0.7; 20.5)
Middle constrictor muscle (MCM)	47.2 ± 9.3	59.3 ± 3.0	0.000	13.1 (−0.7; 23.3)
Inferior constrictor muscle (ICM)	35.9 ± 9.5	53.6 ± 5.0	0.000	18.2 (2.3; 32.3)
Constrictor muscles	48.8 ± 6.6	59.4 ± 2.2	0.000	10.7 (2.2; 20.9)
Musculus cricopharyngeus	33.6 ± 8.6	48.7 ± 7.6	0.000	15.5 (0.9; 29.9)
Esophagus inlet muscle (EIM)	37.2 ± 11.1	45.5 ± 10.0	0.000	6.9 (-1.4; 26.5)
SCM to EIM	46.3 ± 6.4	57.2 ± 2.9	0.000	10.5 (2.7; 20.9)
Cervical esophagus	29.1 ± 8.9	33.5 ± 6.2	0.002	3.5 (−8.3; 19.7)
Soft palate	59.1 ± 5.1	62.0 ± 2.0	0.002	1.0 (-1.9; 13.9)
Base of tongue	55.6 ± 7.7	60.7 ± 2.3	0.000	3.8 (−2.1; 19.9)
Oral cavity	52.2 ± 8.5	57.1 ± 4.5	0.000	5.7 (−2.7; 16.2)
Glottic larynx	30.5 ± 11.4	49.4 ± 8.5	0.000	17.2 (1.5; 35.6)
Supraglottic larynx	41.7 ± 11.2	57.0 ± 4.2	0.000	16.4 (−0.2; 26.4)
Arytenoid cartilages	30.4 ± 10.9	50.1 ± 7.8	0.000	19.0 (2.3; 34.8)
Ipsilateral parotid gland	38.9 ± 7.7	40.1 ± 7.5	0.002	1.5 (−3.6; 3.9)
Contralateral parotid gland	7.4 ± 2.2	21.5 ± 2.8	0.000	15.1 (6.6; 20.4)
Total parotid gland	23.5 ± 4.9	31.1 ± 4.8	0.000	8.0 (3.0; 12.2)
Ipsilateral submandibular gland	60.7 ± 3.7	62.3 ± 5.6	0.088	0.2 (−1.1; 9.2)
Contralateral submandibular gland	17.8 ± 13.8	53.7 ± 2.8	0.000	41.5 (11.4; 45.0)
Total submandibular gland	38.4 ± 7.8	57.9 ± 2.0	0.000	21.7 (5.7; 28.7)

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
