# Peer review of "Dose Reduction to the Swallowing Apparatus and the Salivary Glands by De-Intensification of Postoperative Radiotherapy in Patients with Head and Neck Cancer: First (Treatment Planning) Results of the Prospective Multicenter DIREKHT Trial"

_cancers, 2020, doi:10.3390/cancers12030538_

Round 1
Reviewer 1 Report
The authors addressed all issues.
Although they tried to explain the justification of only enrolling 30 patients it still remains unclear why not all 100 patients were analyzed that have been included so far.
There are still spelling mistakes, e.g. in the title (appartus instead of apparatus) making it unlikely that the manuscript has been thoroughly checked by a language editor.
However, the major concern still exists about the statistics and significance calculations. With an n=30 and high standard deviations the p-values are not plausible. As one of numerous examples: ipsilateral parotid gland with a mean of 38.9+/-7.7 in the study group vs. 40.1+/-7.5 in the standard treatment group showing a p-value of 0.002. The authors referred to another publication (e.g. Stromberger, Budach, Marnitz et al, Unilateral and bilateral neck SIB for head and neck cancer patients, 2016 ) stating that these authors used the same statistical test and showed similar p-values. However, these authors report mainly no significance in their calculations (e.g. also organs at risk parotid and submandibular glands) although showing considerably smaller standard deviations than the current study.
Therefore the decision stands and we recommend rejection of publication.
Reviewer 2 Report
The limitation of the study which I had commented previously, is now clearly addressed in the discussion section. There is a spelling mistake in the title; 'appartus' should be 'apparatus.'
Author Response
Reviewer 2
Dear referee,
Thank you again for revising our manuscript and helping us to improve it.
Comment to the author:
The limitation of the study which I had commented previously, is now clearly addressed in the discussion section. There is a spelling mistake in the title; 'appartus' should be 'apparatus
Thanks. We corrected the mentioned spelling mistake in the title.

Reviewer 3 Report
Almost all my previous comments have been taken into account in order to improve the mansucript.
There is one minor aspect, that Needs to be considered. On page 9 in line 233-235 the authors write "According to reported dose volume relations....regarding side effects like xerostomia or dysphaigia." I certainly miss a reference in this context. It is essential to the reader to provide a reference here! Please add this information.
Author Response
Dear referee,
Thank you again for revising our manuscript and helping us to improve it.
Comments to the author:
Almost all my previous comments have been taken into account in order to improve the mansucript.
There is one minor aspect, that Needs to be considered. On page 9 in line 233-235 the authors write "According to reported dose volume relations....regarding side effects like xerostomia or dysphaigia." I certainly miss a reference in this context. It is essential to the reader to provide a reference here! Please add this information.
Thanks. We added the references in the text.
This manuscript is a resubmission of an earlier submission. The following is a list of the peer review reports and author responses from that submission.
Round 1
Reviewer 1 Report
It remains unclear why only 30 patients are analyzed while more than 100 have been included.
The significance calculations seem erroneous. With an n=30 and high standard deviations the p-values are not comprehensible.
The manuscript still contains spelling and formulation errors. The numbering of the tables is wrong. Abbreviations in the text are not explained.
In the results, patient characteristics are inserted in a table and then repeated unnecessarily in the text.
Reviewer 2 Report
The authors revised the manuscript.
The article is well organized and include valuable findings in the dosimetric analysis of DIREKHT trial, which is ongoing. The dose to most organs at risk were reduced significantly compared to when treated with conventional radiotherapy field.
However, there is still a critical point which need to be addressed.Because there was no patient treated in arm 1, it is apparent that the dosimetric results do not represent those of the entire cohort of the current trial. Planning based on the protocol of arm 1 on dummy patients or actual patients who might have been accrued thereafter, if any, should be considered. At least, this point should be mentioned in the discussion.
Reviewer 3 Report
The manuscript is in essence a treatment planning study linked to the DIREKHT trial. In the treatment planning comparison 30 patients were included. As such I do not find this study very enlightening, because many treatment planning studies have been published during the past two decades. When looking at the conclusion of the manuscript on page 12, there is literally no novel finding. Moreover, such a conclusion can be already drawn from the study design with two very different target concepts. The only merit I see in this publication is the fact that it quantifies the obvious qualitatively known dosimetric outcome due the intrinsic study design. In this respect I even miss an enthusiastic discussion that correlates the dosimetric gain with known or estimated dose response relations, in order be able to discuss the potential clinical impact of the trial.
Another main concern against publication is the rather confusing terminology used in the context of treatment planning, i.e. the authors write about a “conventional treatment plan” and a “protocol-derived treatment plan”. The term “conventional treatment plan” is misleading, as in many other treatment planning studies an advanced technique (VMAT vs 3DCRT) is compared with a conventional technique, or novel beam quality is compared with a conventional one (e.g. protons vs. photons). However, in this study and in both planning situations the same advanced treatment technique of rotational IMRT/VMAT based on computerized treatment plan optimization was applied, so it was no comparison of treatment techniques. Instead the study compared two different target volume concepts, i.e. a protocol derived modified target concept was applied, then a treatment plan with a precision VMAT technique was then generated and compared with the respective dose distribution obtained for the conventional target concept, again using the same treatment technique. The authors specify a 3 to 5 mm margin around the CTV. Here I would like to know on which immobilization and position verification concepts this margin is based on. In Figure 3 the legend to this isodose levels are much too small, and the legends w.r.t. the DVH are not explained at all. So Figure 3 is not self-explanatory and self-standing, respectively. Finally, the manuscript needs thorough language revision.
In summary, the manuscript does to contain too novel information from the radiotherapy point of view, besides the fact that it seems to promote the DIREKHT trial.